# Numerical Assessment of Dipole Interaction with the Single-Phase Nanofluid Flow in an Enclosure: A Pseudo-Transient Approach

**DOI:** 10.3390/ma15082761

**Published:** 2022-04-09

**Authors:** Rashid Ayub, Shabbir Ahmad, Sohail Ahmad, Yasmeen Akhtar, Mohammad Mahtab Alam, Omar Mahmoud

**Affiliations:** 1Centre for Advanced Studies in Pure and Applied Mathematics, Bahauddin Zakariya University, Multan 60800, Pakistan; rashidayub2115@gmail.com; 2Institute of Geophysics and Geomatics, China University of Geosciences, Wuhan 430074, China; shabbiraleem@cug.edu.cn; 3College of Pharmaceutical Sciences, Zhejiang University, Hangzhou 310030, China; dryasmeenakhtar@yahoo.com; 4Department of Basic Medical Sciences, College of Applied Medical Science, King Khalid University, Abha 61421, Saudi Arabia; mmalam@kku.edu.sa; 5Petroleum Engineering, Faculty of Engineering and Technology, Future University in Egypt, New Cairo 11845, Egypt; omar.saad@fue.edu.eg

**Keywords:** dipole, magnetic field, nanofluids, enclosure, pseudo-transient approach

## Abstract

Nanofluids substantially enhance the physical and thermal characteristics of the base or conducting fluids specifically when interacting with the magnetic field. Several engineering processes like geothermal energy extraction, metal casting, nuclear reactor coolers, nuclear fusion, magnetohydrodynamics flow meters, petrochemicals, and pumps incorporate magnetic field interaction with the nanofluids. On the other hand, an enhancement in heat transfer due to nanofluids is essentially required in various thermal systems. The goal of this study is to figure out that how much a magnetic field affects nanofluid flow in an enclosure because of a dipole. The nanofluid is characterized using a single-phase model, and the governing partial differential equations are computed numerically. A Pseudo time based numerical algorithm is developed to numerically solve the problem. It can be deduced that the Reynolds number and the magnetic parameter have a low effect on the Nusselt number and skin friction. The Nusselt number rises near the dipole location because of an increase in the magnetic parameter Mn and the Reynolds number Re. The imposed magnetic field alters the region of high temperature nearby the dipole, while newly generated vortices rotate in alternate directions. Furthermore, nanoparticle volume fraction causes a slight change in the skin friction while it marginally reduces the Nusselt number.

## 1. Introduction

Nanofluids are very eminent in many energy systems and play a crucial role in the heat transfer mechanism. Usual base fluids like alcohol, graphene, engine oil, and water possess a low competency to improve the heat transport rate. Therefore, the amalgamation of base fluid in the tiny sized solid constituents would improve the heat transfer rate [1]. Ultimately, Choi [2] suggested that the tendency of conducting fluids to enhance the thermal properties could be enhanced by mixing the solid fragments (nanoparticles) into base fluids. The diffusion of small nanoparticles sized (1–100 nm) in the base liquids refers to nanofluids. The elements of nanoparticles comprise oxides, carbides, nitrides, and metals such as $SiO_{2}$, CuO, SiN, SiC, Au, Fe, and Cu. With improved and enhanced thermal mechanisms, nanofluids have enormous employments involving thermal storage capacity, nuclear system cooling, vehicle engine cooling, welding cooling, high power lasers, and are used in supersonic and ultrasonic fields, gas recovery of boiler exhaust fuel, tumor and cancer therapy, pharmacology, refrigerator, drag reduction, grinder machines, hybrid power engines, fuel cells, car AC, microelectronic chips, and microwave tubes [3]. Various recent investigations regarding nanofluids subject to various conditions have been studied in [4,5,6,7,8,9,10].

Flows involving magnetohydrodynamic effects have potential uses in fiber coating, plasma confinement, magnetic material processing industries, photochemical reactors, purification of crude oil, magnetic drug targeting, nuclear fusion, and many other fields. Investigations concerning MHD flows have been carried out by many researchers. Yang et al. [11] used the theoretical and experimental results in order to measure the nanowire motion in the presence of a magnetic field. They utilized the MRF (multiple reference frame) model to find the torque of fluid, which was then equated with the theoretical and experimental results. The impact of magnetic nanofluids (MNFs) on the heat transfer enhancement was determined by Zhang and Zhang [12]. Their outcomes depicted that non-magnetic and unidirectional fields inserted a low effect on the coefficient of local heat transfer compared to an alternating magnetic field. Ali et al. [13] studied the flow of aluminum oxide and copper under the action of the Lorentz force. The simulation results were correlated with the experimental ones and were found to be in a good connection with each other. Tetuko et al. [14] examined the heat transmission rate of magnetic nanofluids in a pipe by using different Reynolds numbers, e.g., Re = 285, Re = 228, and Re = 171. The Williamson’s nanofluid under the aligned magnetic field effect was explored numerically by Srinivasulu and Goud [15]. In this problem, the transformation of similarity was adopted to dimensionless the governing equations. An analysis of Casson nanofluid involving magnetohydrodynamic effects verified that the activation energy caused an enhancement in concentration, whereas the radiation effect boosted up the temperature and entropy generation as well (see Shah et al. [16]). The heat transportation flow of micropolar fluid inside a permeable channel, under the magnetohydrodynamic effect, was investigated by Ahmad et al. [17]. The effect of imposed magnetic field not only enhanced the Nusselt number and skin friction, but also caused a rapid increase in the rotation of micro particles. The implicit finite difference based numerical technique, known as Keller–Box, was utilized by Haq et al. [18] to numerically solve the dynamical model of MHD Eyring Powell nanofluid flow. Both the heat transfer rate and surface drag were increased by the curvature parameter and the Eyring Powell fluid parameter.

Nanofluid flows through cavities with heat transfer characteristics under the magnetohydrodynamic effects have been extensively studied by numerous researchers because of their practical implementations in several areas of science and technology. MHD nanoparticles flow inside a lid driven cavity, under the impact convective conditions on boundary, was discussed numerically by Muthtamilselvan and Doh [19]. The walls of the cavity were located at different positions. Moreover, the bottom and horizontal cavity walls possessed different temperatures, while the right and left cavity walls were considered to be insulated. Giwa et al. [20] expressed the importance of magnetohydrodynamic phenomena in cavities filled with nanofluids. They proposed that the natural convection flows of nanofluids together with heat transfer in cavities could maintain and control the thermal efficiency of fluid using magnetic field sources, porous media, and aspect ratio. A simulation analysis was performed by Selimefendigil and Öztop [21] to interpret the CNT–water nanofluid flow within a corrugated square cavity under an inclined magnetic field environment. The two cavity walls were assumed to be adiabatic and the other two were kept at same temperatures. They adopted the residual Galerkin weighted technique to find the numerical solution. Geridonmez and Oztop [22] offered a novel numerical study regarding the flow of aluminum oxide nanoparticles inside a cavity subject to natural convection and cross partial magnetic fields. The radial basis functions (Rbfs) were introduced to solve the governing vorticity and stream function equations.

The cavity flows have considerable effects on the heat transportation and thermal performance of many systems, and play an essential role in cooling and heating energy systems. Examples incorporate solar collectors, nuclear reactors, energy storage geothermal reservoirs, underground water flow, and boilers. Flows of different fluids within a cavity have been analyzed by several researchers. The flow of ferrofluid (Fe_3_O_4_) together with H_2_O in a cavity, under the entropy generation effect, was carried out by Mohammadpourfard et al. [23]. In this numerical analysis, the control volume method was utilized to solve the two-phase mixture model. The effects produced by the buoyancy force and non-uniform magnetic field on the heat transfer flow in the cavity were investigated. Revnic et al. [24] considered the flow inside a cavity filled with aluminum oxide and copper nanoparticles. A high temperature was taken on the two walls of the cavity while the other two were thermally insulated. Heat transfer enhancement was noted in the cavity for hybrid nanoparticles. The flow within a square enclosure was taken by Yasmin et al. [25] to analyze the heat transport and flow features of MHD Cassonna nanofluids. Further investigations on cavity flows can be found in [26,27,28].

The aforementioned literature review evidently discloses that the effects of moving diploe on the nanofluid flow occurring in a cavity have not been numerically interpreted yet. The recent work describes the novel aspects of the problem under the magnetohydrodynamic effects. The single-phase model (SPM) has been adopted to simplify the nanofluid flow model. The tabular and graphical outcomes express the dominant effects of the preeminent parameters on the heat transfer and flow in the cavity filled with nanofluids. New vortices seem to be appearing in the flow field with the existence of a dipole in the vicinity of cavity.

## 2. Materials and Methods

Figure 1 depicts a schematic geometry of the problem under consideration. A square with side L acts as the computational domain. Because of the mechanical arrangement, the top lid moves in the right side. Temperatures $T_{h}$ and $T_{c}$ show upper and lower wall temperatures, respectively, which are kept constant, and a magnetic source is placed at (a,b)=(L/2,−0.05). The water is considered to be the base fluid, while the solid particles of aluminum oxides Al_2_O_3_ are considered nanoparticles, which have been used in the nanofluid.

To examine the thermodynamic properties and flow characteristics of an incompressible and laminar nanofluid flow under the dipole influence, we formulated the governing equations using SPM, as follows:(1)$$\frac{\partial \tilde{U}}{\partial X}+\frac{\partial \tilde{V}}{\partial Y}=0,$$
(2)$$\frac{\partial \tilde{U}}{\partial t^{\prime }}+\left(\tilde{U}\frac{\partial \tilde{U}}{\partial X}+\tilde{V}\frac{\partial \tilde{U}}{\partial Y}\right)=-\frac{1}{{\tilde{\rho }}_{nf}}\frac{\partial P}{\partial X}+{\tilde{\upsilon }}_{nf}\left(\frac{\partial^{2}\tilde{U}}{\partial X^{2}}+\frac{\partial^{2}\tilde{U}}{\partial Y^{2}}\right)+\frac{{\tilde{\mu }}_{o}\tilde{M}}{{\tilde{\rho }}_{nf}}\frac{\partial \tilde{H}}{\partial X},$$
(3)$$\frac{\partial \tilde{V}}{\partial t^{\prime }}+\left(\tilde{U}\frac{\partial \tilde{V}}{\partial X}+\tilde{V}\frac{\partial \tilde{V}}{\partial Y}\right)=-\frac{1}{{\tilde{\rho }}_{nf}}\frac{\partial P}{\partial Y}+{\tilde{\upsilon }}_{nf}\left(\frac{\partial^{2}\tilde{V}}{\partial X^{2}}+\frac{\partial^{2}\tilde{V}}{\partial Y^{2}}\right)+\frac{{\bar{\mu }}_{o}\tilde{M}}{{\tilde{\rho }}_{nf}}\frac{\partial \tilde{H}}{\partial Y},$$
(4)$$\begin{array}{l} \frac{{\left(\tilde{\rho }c_{p}\right)}_{nf}}{k_{nf}}\left(\tilde{U}\frac{\partial T}{\partial X}+\tilde{V}\frac{\partial T}{\partial Y}\right)+\left(\frac{{\bar{\mu }}_{o}}{k_{nf}}\right)T\frac{\partial \tilde{M}}{\partial T}\left(\tilde{U}\frac{\partial \tilde{H}}{\partial X}+\tilde{V}\frac{\partial \tilde{H}}{\partial Y}\right)\\ =\nabla^{2}T+\left(\frac{{\tilde{\mu }}_{nf}}{k_{nf}}\right)\left\{2{\left(\frac{\partial \tilde{U}}{\partial X}\right)}^{2}+2{\left(\frac{\partial \tilde{V}}{\partial Y}\right)}^{2}+{\left(\frac{\partial \tilde{V}}{\partial X}+\frac{\partial \tilde{U}}{\partial Y}\right)}^{2}\right\}. \end{array}$$

Here:

${\bar{\mu }}_{0}\tilde{M}\frac{\partial \tilde{H}}{\partial X}\;$ and ${\bar{\mu }}_{0}\tilde{M}\frac{\partial \tilde{H}}{\partial Y}\;$ stands for the magnetic force components along *x* and *y*-axis, respectively.

${\bar{\mu }}_{0}T\frac{\partial \tilde{M}}{\partial T}(\tilde{U}\frac{\partial \tilde{H}}{\partial X}+\tilde{V}\frac{\partial \tilde{H}}{\partial Y})$ expresses the magneto-caloric phenomenon subject to the thermal power per unit volume.

Where $\tilde{H}$ is the magnetic field intensity attributed to the existence of a dipole at (a,b).

$\tilde{H}(X,Y)=\frac{\gamma }{2\pi }\frac{\left(X-a\right)}{{\left(X-a\right)}^{2}+{\left(Y-b\right)}^{2}}$, with γ being the strength of the magnetic field at the dipole location [29].

$\tilde{M}$ represents the magnetization property, which is calculated as a function of magnetic field strength and the temperature of the fluid. The following is a simple linear relationship for $\tilde{M}$:$$\tilde{M}=K\tilde{H}\left({\bar{T}}_{c}-T\right).$$
where K is a pyro magnetic factor, while ${\bar{T}}_{c}$ represents the Curie temperature [30].All other terms have their conventional meanings.

It is worth noting that the physical characteristics of the nanofluid are denoted by the subscript (nf).

Boundary conditions:(5)$$\begin{array}{l} \tilde{U}(X,\;0)=V_{o},\;\tilde{U}(X,\;L)=V_{o},\;\tilde{V}(X,\;L)=0,\;\;\;\;\;\tilde{V}(X,\;0)=0,\;T(X,\;0)=T_{c},\;T(X,\;L)=T_{h};\;\;0<X<L.\\ \tilde{V}(0,Y)=\tilde{U}(0,Y)=0,\;\;\tilde{V}(L,Y)=\tilde{U}(L,Y)=0,\;{\left(\frac{\partial T}{\partial X}\right)}_{X=0}=0,\;{\left(\frac{\partial T}{\partial X}\right)}_{X=L}=0;\;\;\;\;\;0<Y<L. \end{array}$$

After removing the pressure term, we get the following:(6)$$\begin{array}{l} \frac{\partial }{\partial t^{\prime }}\left(\frac{\partial \tilde{U}}{\partial Y}-\frac{\partial \tilde{V}}{\partial X}\right)+\tilde{U}\frac{\partial }{\partial X}\left(\frac{\partial \tilde{U}}{\partial Y}-\frac{\partial \tilde{V}}{\partial X}\right)+\tilde{V}\frac{\partial }{\partial Y}\left(\frac{\partial \tilde{U}}{\partial Y}-\frac{\partial \tilde{V}}{\partial X}\right)\\ ={\tilde{\upsilon }}_{nf}\left(\frac{\partial^{2}}{\partial X^{2}}+\frac{\partial^{2}}{\partial Y^{2}}\right)\left(\frac{\partial \tilde{U}}{\partial Y}-\frac{\partial \tilde{V}}{\partial X}\right)+\left(\frac{\partial \left(\frac{{\bar{\mu }}_{o}\tilde{M}}{{\tilde{\rho }}_{nf}}\frac{\partial \tilde{H}}{\partial X}\right)}{\partial Y}-\frac{\partial \left(\frac{{\bar{\mu }}_{o}\tilde{M}}{{\tilde{\rho }}_{nf}}\frac{\partial \tilde{H}}{\partial Y}\right)}{\partial X}\right). \end{array}$$

We use the dimensionless variables listed below:(7)$$x=\frac{X}{\tilde{L}},\;\;y=\frac{Y}{\tilde{L}},\;\;t=\frac{v_{0}}{\tilde{L}}t^{\prime },\;\;u=\frac{\tilde{U}}{v_{o}},\;\;\tilde{\theta }=\frac{T-T_{c}}{\Delta T},\;v=\frac{\tilde{V}}{v_{o}},\;$$

Now, Equations (4) and (6) imply that:(8)∂ω˜∂t+u∂ω˜∂x+v∂ω˜∂y=(1−φ˜+φ˜ρ˜sρ˜f306φ˜2−0.19φ˜+1)−11Re∇2ω˜+(Mn1−φ˜+φ˜ρ˜sρ˜f) H˜ (∂H˜∂y.∂θ˜∂x−∂H˜∂x.∂θ˜∂y),
(9)∇2θ˜=Pr(1−φ˜+φ˜(ρ˜cp)s(ρ˜cp)f28.905φ˜2+2.8273φ˜+1)Re(306φ˜2−0.19φ˜+11−φ˜+φ˜ρ˜sρ˜f){∂θ∂x∂ψ˜∂y−∂θ∂y∂ψ˜∂x}+MnPr(1−φ˜+φ˜(ρ˜cp)s(ρ˜cp)f28.905φ˜2+2.8273φ˜+1)Re(306φ˜2−0.19φ˜+1(1−φ˜+φ˜ρ˜sρ˜f)2)EcH˜(ε−ψ˜){∂H˜∂x∂ψ˜∂y−∂H˜∂y∂ψ˜∂x}+Pr(1−φ˜+φ˜(ρ˜cp)s(ρ˜cp)f28.905φ˜2+2.8273φ˜+1)Ec{(∂2ψ˜∂y2−∂2ψ˜∂x2)2+4(∂2ψ˜∂x∂y)2},

The dimensionless parameters are portrayed in Table 1.The above equations signify stream function–vorticity form, which is the modified version of the Equations (1)–(4), with the following:(10)$$\tilde{u}=\frac{\partial \tilde{\psi }}{\partial y},\;\;\tilde{v}=\frac{\partial \tilde{\psi }}{\partial x}\;\mathrm{and}\;\;\left(\frac{\partial \tilde{u}}{\partial y}-\frac{\partial \tilde{v}}{\partial x}\right)=-\tilde{\omega }\;\mathrm{or}\;\left\{\left(\frac{\partial^{2}\tilde{\psi }}{\partial x^{2}}+\frac{\partial^{2}\tilde{\psi }}{\partial y^{2}}\right)=-\tilde{\omega }\right\}.$$

It is essential to state that the physical properties of the nanofluid (with *nf* subscripts) given in Equations (1)–(4) will be analyzed using the relations described in [31].

For example,
$${\left(\tilde{\rho }c_{p}\right)}_{nf}=(1-\tilde{\varphi }){\left(\tilde{\rho }c_{p}\right)}_{f}+(\tilde{\rho }c_{p})\tilde{\varphi },\;\;\alpha_{nf}=\frac{k_{nf}}{{\left(\tilde{\rho }c_{p}\right)}_{nf}},\;\;\frac{k_{nf}}{k_{f}}=\frac{k_{s}-2\tilde{\varphi }(k_{f}-k_{s})+2k_{f}}{k_{s}+\tilde{\varphi }(k_{f}-k_{s})+2k_{f}}.$$

In the same way, the boundary conditions take the following form:(11)$$\begin{array}{l} \tilde{v}(x,0)=0,\;\;\tilde{v}(x,1)=0,\;\;\;\;\tilde{u}(x,0)=0,\;\tilde{u}(x,1)=1,\;\tilde{\theta }(x,0)=0,\;\tilde{\theta }(x,1)=1;\;\;\;\;\;\;\;\;\;0<x<1\\ \tilde{v}(0,y)=\tilde{u}(0,y)=0,\;\;\tilde{v}(1,y)=\tilde{u}(1,y)=0,\;{\left(\frac{\partial \tilde{\theta }}{\partial x}\right)}_{x=0}=0,\;{\left(\frac{\partial \tilde{\theta }}{\partial x}\right)}_{x=1}=0;\;\;\;\;\;0<y<1. \end{array}$$

## 3. Numerical Approach

The dimensionless coupled Equations (8)–(10) have been iteratively solved with respect to the boundary conditions given in Equation (11). A pseudo-transient approach, which involves time as an iteration parameter, is employed to determine the numerical solution of the problem.

### 3.1. Numerical Solution

For the numerical solution, we may incorporate central differences for the spatial derivatives, whereas a predictor–corrector-like approach may be followed for the time integration. With the superscripts representing the time level and the subscripts denoting the location of the grid point, the system of governing equations when discretized, as stated above, may be written as follows:(12)$$\frac{{\tilde{\psi }}_{i-1,j}^{\left(n+1/2\right)}-2{\tilde{\psi }}_{i,j}^{\left(n+1/2\right)}+{\tilde{\psi }}_{i+1,j}^{\left(n+1/2\right)}}{h^{2}}+\frac{{\tilde{\psi }}_{i,j-1}^{\left(n+1/2\right)}-2{\tilde{\psi }}_{i,j}^{\left(n+1/2\right)}+{\tilde{\psi }}_{i,j+1}^{\left(n+1/2\right)}}{k^{2}}=-{\tilde{w}}_{i,j}^{\left(n\right)}$$
(13)$${\tilde{u}}_{i,j}^{\left(n+1/2\right)}=\frac{{\tilde{\psi }}_{i,j+1}^{\left(n+1/2\right)}-{\tilde{\psi }}_{i,j-1}^{\left(n+1/2\right)}}{2k}$$
(14)$${\tilde{v}}_{i,j}^{\left(n+1/2\right)}=-\frac{{\tilde{\psi }}_{i+1,j}^{\left(n+1/2\right)}-{\tilde{\psi }}_{i-1,j}^{\left(n+1/2\right)}}{2h}$$
(15)w˜i,j(n+1/2)−w˜i,j(n)δt2=(1−φ˜+φ˜ρsρf306φ˜2−0.19φ˜+1)−11Re{−2w˜i,j(n+1/2)+w˜i−1,j(n+1/2)+w˜i+1,j(n+1/2)h2+−2w˜i,j(n)+w˜i,j−1(n)+w˜i,j+1(n)k2}+(Mn1−φ˜+φ˜ρsρf)H˜i,j{−H˜i,j−1+H˜i,j+12k−θ˜i−1,j(n)+θ˜i+1,j(n)2h−−H˜i−1,j+H˜i+1,j2h−θ˜i,j−1(n)+θ˜i,j+1(n)2k}−(w˜i+1,j(n+1)−w˜i−1,j(n+1)2h)u˜i,j(n+1/2)−v˜i,j(n+1/2)(w˜i,j+1(n)−w˜i,j−1(n)2k)
(16)θ˜i,j(n+1/2)−θ˜i,j(n)δt2=(28.905φ˜2+2.8273φ˜+11−φ˜+φ˜(ρcp)s(ρcp)f)(1−φ˜+φ˜ρsρf306φ˜2−0.19φ˜+1){θ˜i−1,j(n+1/2)−2θ˜i,j(n+1/2)+θ˜i+1,j(n+1/2)h2+θ˜i,j−1(n)+θ˜i,j+1(n)−2θ˜i,j(n)k2}+Mn PrEc H˜i,j(ψ˜i,j−ε){u˜i,j(n+1/2)H˜i,j+1−H˜i,j−12k+v˜i,j(n+1/2)H˜i+1,j−H˜i−1,j2h}−u˜i,j(n+1/2)(θ˜i+1,j(n+1/2)−θ˜i−1,j(n+1/2)2h)−v˜i,j(n+1/2)(θ˜i,j+1(n)−θ˜i,j−1(n)2k)−  (1−φ˜+φ˜ρsρf306φ˜2−0.19φ˜+1) EcRe{(u˜i,j+1(n+1/2)−u˜i,j−1(n+1/2)2k+v˜i+1,j(n+1/2)−v˜i−1,j(n+1/2)2h)2+4(u˜i+1,j(n+1/2)−u˜i−1,j(n+1/2)2h)2}
(17)$$\frac{{\tilde{\psi }}_{i-1,j}^{\left(n+1\right)}+{\tilde{\psi }}_{i+1,j}^{\left(n+1\right)}-2{\tilde{\psi }}_{i,j}^{\left(n+1\right)}}{h^{2}}+\frac{{\tilde{\psi }}_{i,j-1}^{\left(n+1\right)}+{\tilde{\psi }}_{i,j+1}^{\left(n+1\right)}-2{\tilde{\psi }}_{i,j}^{\left(n+1\right)}}{k^{2}}=-{\tilde{w}}_{i,j}^{\left(n+1/2\right)}$$
(18)$${\tilde{u}}_{i,j}^{\left(n+1\right)}=\frac{-{\tilde{\psi }}_{i,j-1}^{\left(n+1\right)}+{\tilde{\psi }}_{i,j+1}^{\left(n+1\right)}}{2k}$$
(19)$${\tilde{v}}_{i,j}^{\left(n+1\right)}=-\frac{{\tilde{\psi }}_{i+1,j}^{\left(n+1\right)}-{\tilde{\psi }}_{i-1,j}^{\left(n+1\right)}}{2h}$$

Finally, we have to numerically solve the following equations in order to determine the numerical solution of the problem:(20)w˜i,j(n+1)−w˜i,j(n+1/2)δt2=(1−φ˜+φ˜ρsρf306φ˜2−0.19φ˜+1)−11Re{w˜i−1,j(n+1/2)−2w˜i,j(n+1/2)+w˜i+1,j(n+1/2)h2+w˜i,j−1(n+1)−2w˜i,j(n+1)+w˜i,j+1(n+1)k2}+(Mn1−φ˜+φ˜ρsρf)H˜i,j{H˜i,j+1−H˜i,j−12kθ˜i+1,j(n+1/2)−θ˜i−1,j(n+1/2)2h−H˜i+1,j−H˜i−1,j2hθ˜i,j+1(n+1/2)−θ˜i,j−1(n+1/2)2k}−u˜i,j(n+1)(w˜i+1,j(n+1/2)−w˜i−1,j(n+1/2)2h)−v˜i,j(n+1)(w˜i,j+1(n+1)−w˜i,j−1(n+1)2k)
(21)θ˜i,j(n+1)−θ˜i,j(n+1/2)δt2=(28.905φ˜2+2.8273φ˜+11−φ˜+φ˜(ρcp)s(ρcp)f)(1−φ˜+φ˜ρsρf306φ˜2−0.19φ˜+1){θ˜i−1,j(n+1/2)−2θ˜i,j(n+1/2)+θ˜i+1,j(n+1/2)h2+θ˜i,j−1(n+1)−2θ˜i,j(n+1)+θ˜i,j+1(n+1)k2}+Mn Ec PrH˜i,j(ψ˜i,j(n+1)−ε){u˜i,j(n+1)H˜i,j+1−H˜i,j−12k+v˜i,j(n+1)H˜i+1,j−H˜i−1,j2h}−u˜i,j(n+1)(θ˜i+1,j(n+1/2)−θ˜i−1,j(n+1/2)2h)−v˜i,j(n+1)(θ˜i,j+1(n+1)−θ˜i,j−1(n+1)2k)− (1−φ˜+φ˜ρsρf306φ˜2−0.19φ˜+1) EcRe{(u˜i,j+1(n+1)−u˜i,j−1(n+1)2k+v˜i+1,j(n+1)−v˜i−1,j(n+1)2h)2+4(u˜i+1,j(n+1)−u˜i−1,j(n+1)2h)2}
which describe how the solution is marched from the *n*th time level to the (n + 1) level. Here, *h* and *k* represent the grid-spacing in the horizontal and vertical directions, respectively, where the time-stepping is represented by δt. A line-by-line approach may then be followed to solve the system of algebraic equations by employing some direct or iterative methods. The required steady-state solution is assumed to be reached when the solution at two consecutive time levels differs by some prescribed tolerance. The iterations are stopped if following criteria are fulfilled:$$\left(||u^{\left(k+1\right)}-u^{\left(k\right)}|{|}_{L_{2}}\right)<TOL_{iter},\;\;\;\;\left(||w^{\left(k+1\right)}-w^{\left(k\right)}|{|}_{L_{2}}\right)<TOL_{iter},\;\;\;\left(||\theta^{\left(k+1\right)}-\theta^{\left(k\right)}|{|}_{L_{2}}\right)<TOL_{iter}$$

$TOL_{iter}=10^{-7}$ is fixed in all of the calculations performed here.

### 3.2. Flow Chart for the Pseudo Transient Method

The pseudo transient approach can solve complex dynamical nonlinear problems in an efficient way. Pseudo time is a mathematical time function that accounts for the parameters involved in the dynamical problems. The objective of the pseudo time analysis is to take a collection of parametric data from a dynamical problem (as in the concerned problem) and to provide numerical outcomes. The solution algorithm is implicit pseudo-time dependent for which Equations (20) and (21) are linearized around time-level n + 1 to obtain the numerical solution. In this method, central differences are used to discretize the governing equations. After finding the stream vorticity function, we determine skin friction $C_{f}R_{e}$, Nusselt number $N_{u}$, streamlines, and isotherms. The flow chart of the pseudo transient method is provided in Figure 2.

## 4. Numerical Results and Discussion

To appraise the precision of our numerical procedure, we solved the problem provided by Shih and Tan [32] by developing our code. In [32], it was assumed that the top lid of cavity moved with variable velocity $U_{o}\left(\tilde{x}\right)={\tilde{x}}^{2}-2\;{\tilde{x}}^{3}+{\tilde{x}}^{4}$. Further, the cavity was filled with a classical Newtonian fluid in the absence of any dipole. There is an analytical solution for this problem, in which the velocity distributions are defined by $\tilde{u}\left(\tilde{x},\;\tilde{y}\right)=8\;\left({\tilde{x}}^{2}-2\;{\tilde{x}}^{3}+{\tilde{x}}^{4}\right)\left(4\;{\tilde{y}}^{3}-2\;\tilde{y}\right)$ and $\tilde{v}\left(\tilde{x},\;\tilde{y}\right)=-8\;\left(4{\tilde{x}}^{3}-6\;{\tilde{x}}^{2}+2\tilde{x}\right)\left({\tilde{y}}^{4}-\;{\tilde{y}}^{2}\right)$. The horizontal velocity profile (calculated analytically and numerically) has been compared in Figure 3. 

In addition, the average Nusselt number on the hot wall is correlated with the established results (see in Table 2) developed by Chen et al. [33] and De Vahl Davis [34]. Our numerical approach is validated by an excellent comparison. In addition, Figure 4 depicts the computational grid that is used in this investigation. All numerical simulations are performed on a grid that is uniform with a step size *h* = 0.01.

We will look at the impact of preeminent factors like the Reynolds number Re, the nanoparticle volume fraction $\tilde{\varphi }$, and the magnetic number Mn, with dipole interaction via cavity flow (caused by the upper and lower lids move in opposite directions). Water is considered to be the base fluid, while the only solid particles of Al_2_O_3_ are considered in the nanofluid. Furthermore, in our simulations, we set the value of ε to be 0.10 and $Ra=10^{4}$, while the physical characteristics of water refer to Pr=6.2 (Table 3).

In the present case, the Eckert number is presumed to be quite small (e.g., $10^{-5}$) because of the comparatively smaller value of the Reynolds number. As illustrated in Figure 5, a step size h=0.01 has been used with a uniform grid that has been chosen in the context of the grid independence analysis.

The fundamental physical characteristics of the Nusselt number $\left(N_{u}\right)$ and local skin friction $\left(C_{f}\right)$ are:$$N_{u}=\frac{\tilde{q}L}{k_{nf}\Delta T}$$
and
$$C_{f}=\frac{2\tau }{{\tilde{\rho }}_{nf}v_{o}{}^{2}}$$
where,
$$\tilde{q}={-k_{nf}(\;\frac{\partial T}{\partial Y})|}_{Y=0,L}$$
heat flux
$${\tau ={\tilde{\mu }}_{nf}(\frac{\;\partial \tilde{U}}{\partial Y})|}_{Y=0,L}$$
shear stress.

By using the dimensionless variables, we obtain following relation:$$C_{f}R_{e}=2\frac{\left(306{\tilde{\varphi }}^{2}-0.19\tilde{\varphi }+1\right)}{\left(1-\tilde{\varphi }+\tilde{\varphi }\frac{{\tilde{\rho }}_{s}}{{\tilde{\rho }}_{f}}\right)}\frac{\partial \tilde{u}}{\partial y}$$
and
$$N_{u}=\frac{\partial \tilde{\theta }}{\partial y}.$$

Along the lower edge of cavity, the values of skin friction $C_{f}R_{e}$ and Nusselt number Nu will be taken into account (for which location of the dipole is at x=0.5). From Figure 6, it can be seen that the influence of Reynold number and a magnetic field is likely to be similar for shear stress $C_{f}R_{e}$ and the Nusselt number Nu. The Nusselt number rises near the dipole location due to the increase in magnetic parameter Mn and the Reynold number Re. The pattern of skin friction $C_{f}R_{e}$ is almost symmetric around the origin. We further notice that as Mn and Re rises, the variation in the vorticity profile is steeper, which varies from negative values to positive values, whereas vorticity is zero at the point where the dipole is situated. However, the rotation in the fluid is observed in the positive and negative direction when the dipole moves from left to right. As the Mn and Re increase, the rotation of the vortices in the opposite direction is noticed. Figure 7 clearly shows this phenomenon.

Now we try to grasp the effect of the nanoparticle volume fraction $\tilde{\varphi }$ on the Nusselt number and local skin friction. The change of Nusselt number and local skin fraction is not very sharp along the left and right adiabatic walls. It is noticed that the nanoparticle volume fraction upper the Nusselt number about the middle of the cavity while falls local skin friction around the dipole location.

At the beginning, when there is no dipole located near the cavity, the behavior of the streamlines is quite smooth with a clockwise vortex near the upper wall. As the upper plate is moving, a high-velocity gradient can be seen along the upper wall. An anticlockwise lowering vortex seems to appear near the dipole from right side with the impact of magnetic field. As the magnetic fields strengthen, this lower vortex starts growing, and another tiny vortex appears from the primary vortex. Consequently, there are three vortices in total at Mn=3000, and the two secondary vortices are originating near the dipole location.

The isotherm has a smooth pattern at the time when no magnetic field is applied, and the isotherms possess a region with a higher thermal gradient along the upper side of the left wall. The zone of higher temperature gradient is created by the applied magnetics around the location of the dipole (Figure 8).

Figure 9 depicts that for Re=1, there is a single primary vortex and the direction of streamline from left to right, it is due to the movement of the upper lid from left to rightward, which is why the flow velocity is more enhanced near the top lid. As we increase the Reynold number, another lowering anticlockwise vortex appears near the right wall of the bottom lid. This newly formed vortex gets bigger and bigger near the dipole for higher values of Reynold numbers and it is originates near the dipole.

It is obvious from Figure 10 that the thermal field becomes steeper with increasing values of the Reynold number along the upper horizontal wall, and also creates a higher thermal gradient near the dipole location.

Table 4 shows that the Nusselt number is reduced by approximately 19% as a result of a 10% rise in the volume fraction of nanoparticles, equated to a 4% variation in the skin friction along the lower wall and negligible change in $\;Cf_{u}$. This reveals that $\tilde{\varphi }$ is even more efficient on the Nusselt number.

It can be seen from Table 5 that a significant impact of the Reynolds number is noted for $Nu_{l}$, $Nu_{u}$, and $Cf_{l}$ in comparison to the only 19% change in $Cf_{u}$. The reality of this fact is that an increase in the movement of the surface tends to enhance the Reynolds number for the fixed thermo-physical properties of nanofluids. Due to this phenomenon, heat energy varies near the moving walls.

Table 6 shows that magnifying the magnetic parameter Mn up to 3000 gives rise to $Nu_{l}$ and $Nu_{u}$ by 115% and 110%, respectively, and a remarkable variation is found in the shear stress along the bottom lid of the cavity, while a negligible change is in $Cf_{u}$. This means that the dipole is more efficient for shear stress along the lower wall rather than the upper wall of the cavity, and this happens due to the motion of the upper lid.

## 5. Conclusions

An inclusive computational analysis of the dipole interaction with the flow of nanoparticles (*Al_2_O_3_*) and base fluid (*H_2_O*) within an enclosure is presented in this work. The main purpose of this study is to examine that how much a nanofluid flow is affected by the magnetic field inside an enclosure due to the presence of a dipole. The single-phase model is adopted to characterize the nanofluid, and the governing partial differential equations are computed numerically. The results evidently reveal that 10% increase in the nanoparticles volume fraction $\tilde{\varphi }$ leads to an 19% enhancement in Nusselt number and 4% in the skin friction. Hence, it can be deduced that $\tilde{\varphi }$ is even more efficient on the Nusselt number compared to its effect on skin friction. We noticed the following key points in our analysis:The Reynolds number and the magnetic field give rise to the Nusselt number in the surroundings of the dipole.The new vortices also seem to be rotating in the alternating paths due to dipole movement.The magnetic field and Reynold number form more vertices near the dipole location.The imposed magnetic field shifts the thermal gradient around the dipole location.No significant change is found in the shear stress for $\tilde{\varphi }$, whereas the Nusselt number strictly decreases against the solid nanoparticles volume fraction $\tilde{\varphi }$.The influence of Reynold’s number on the Nusselt number and $Cf_{l}$ is significant compared to $Cf_{u}$.The skin friction becomes more effective along the lower wall in the presence of the dipole.

## Figures and Tables

**Figure 1 materials-15-02761-f001:**
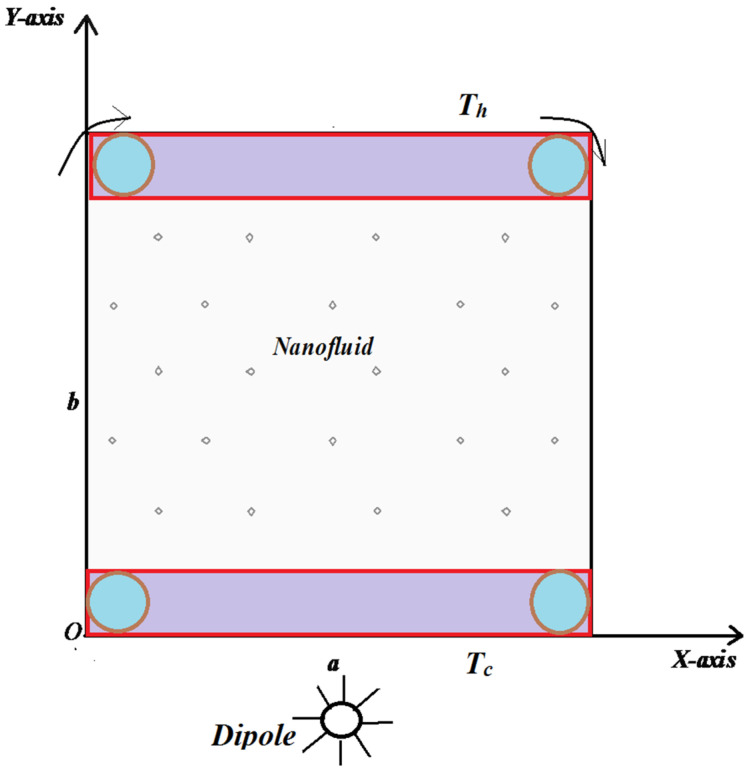
Problem schematic diagram.

**Figure 2 materials-15-02761-f002:**
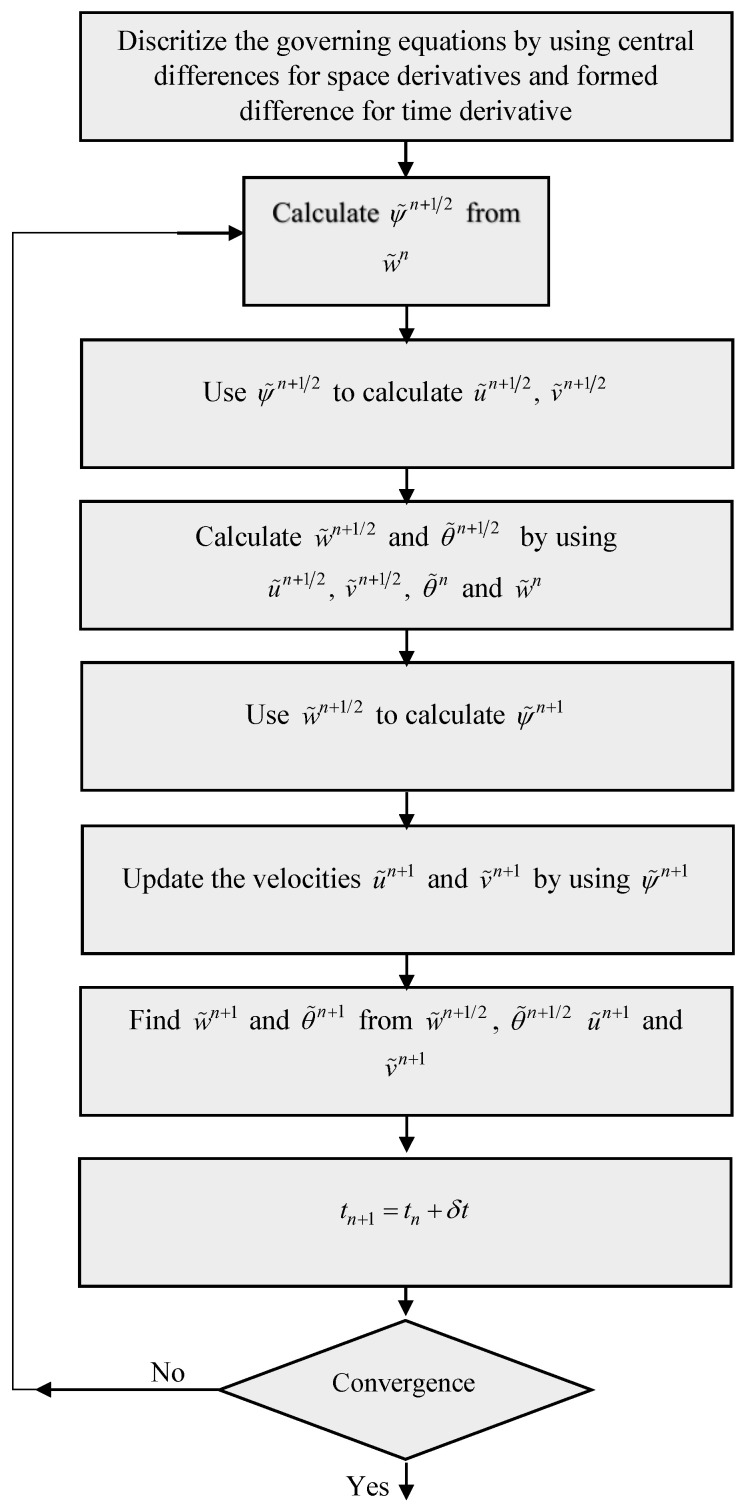
Flow chart diagram for the Pseudo Transient Method.

**Figure 3 materials-15-02761-f003:**
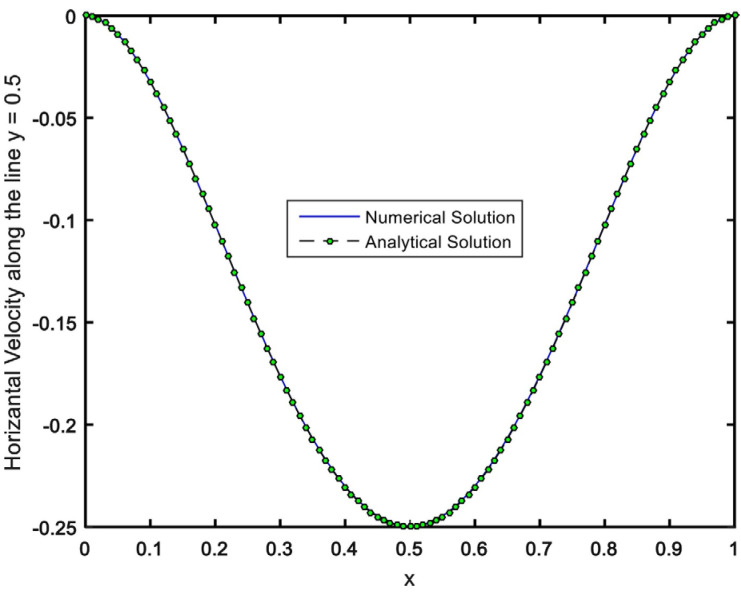
Numerical results comparison with the analytical solution of Shih and Tan [32].

**Figure 4 materials-15-02761-f004:**
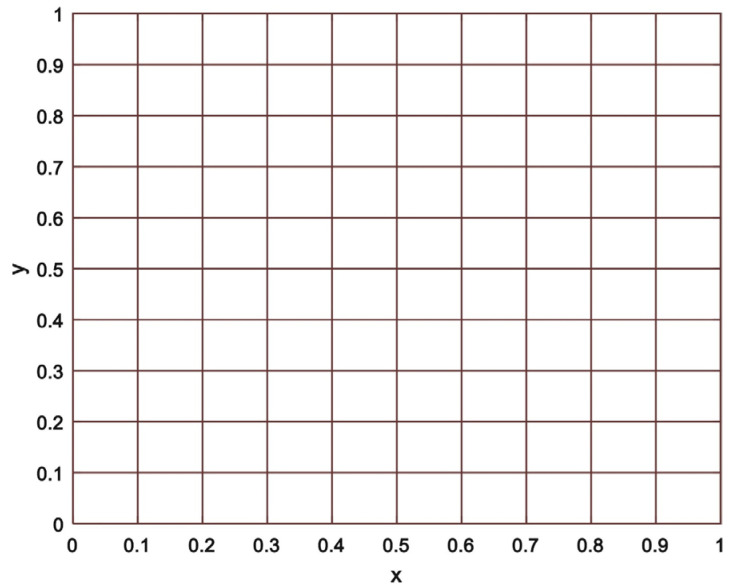
Computational grid of current study.

**Figure 5 materials-15-02761-f005:**
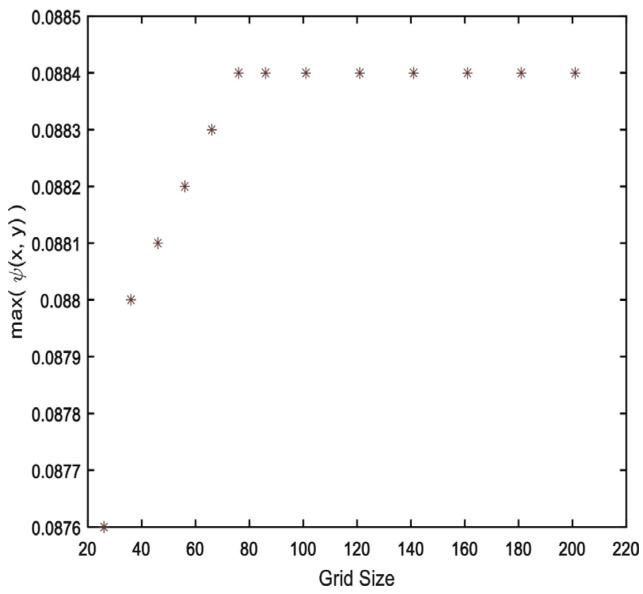
Grid Independence Study.

**Figure 6 materials-15-02761-f006:**
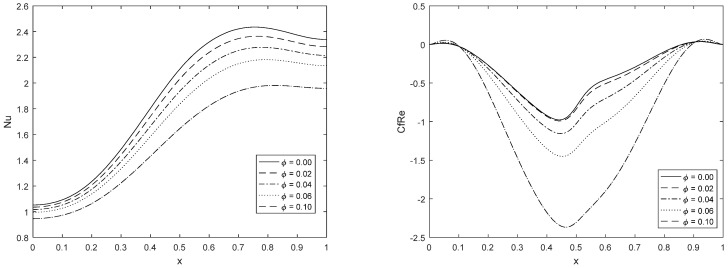
Change in $C_{f}R_{e}$ and $N_{u}$ at the dipole location along the lower wall.

**Figure 7 materials-15-02761-f007:**
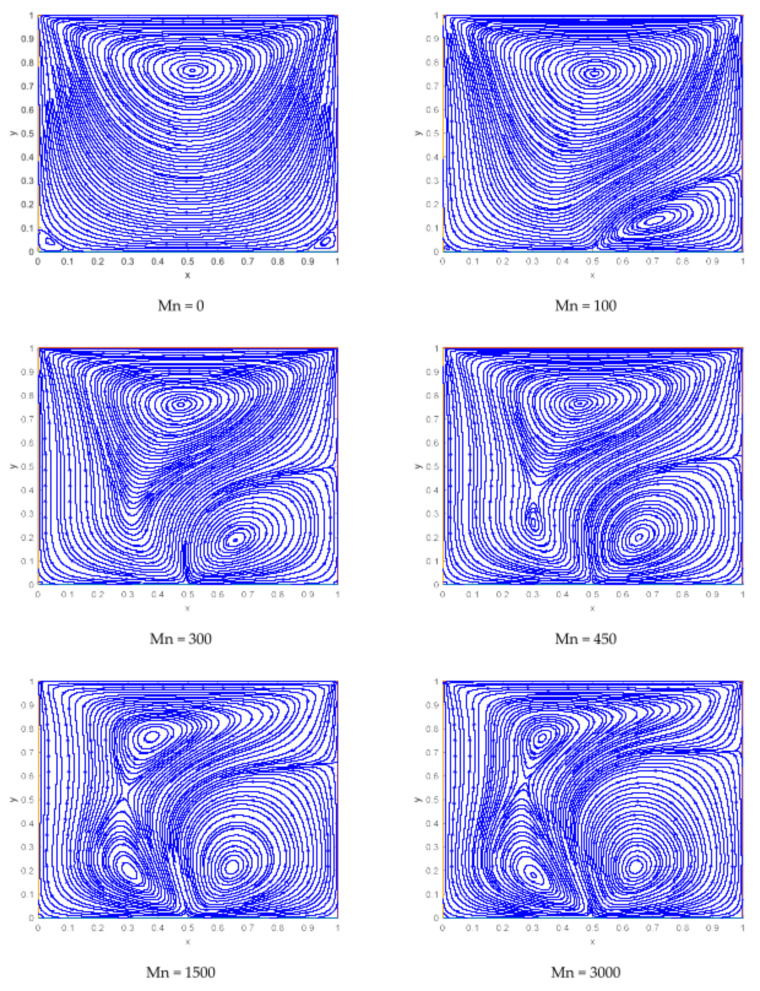
Representation of streamlines for various values of magnetic interaction parameters.

**Figure 8 materials-15-02761-f008:**
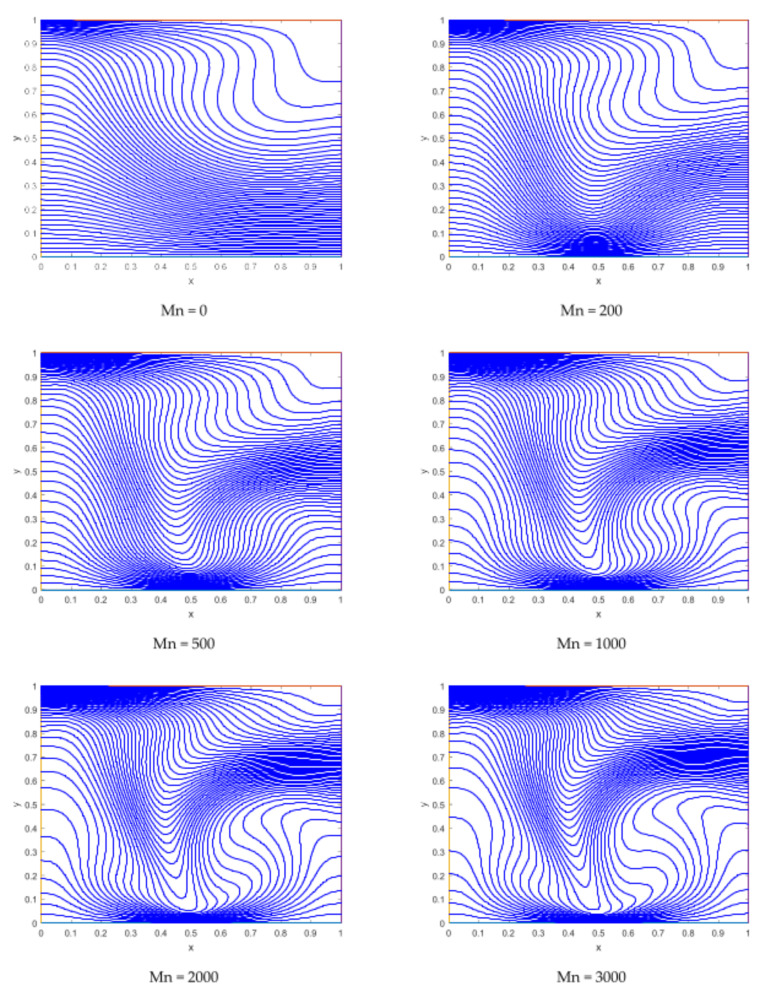
Representation of isotherms for different values of the magnetic interaction parameters.

**Figure 9 materials-15-02761-f009:**
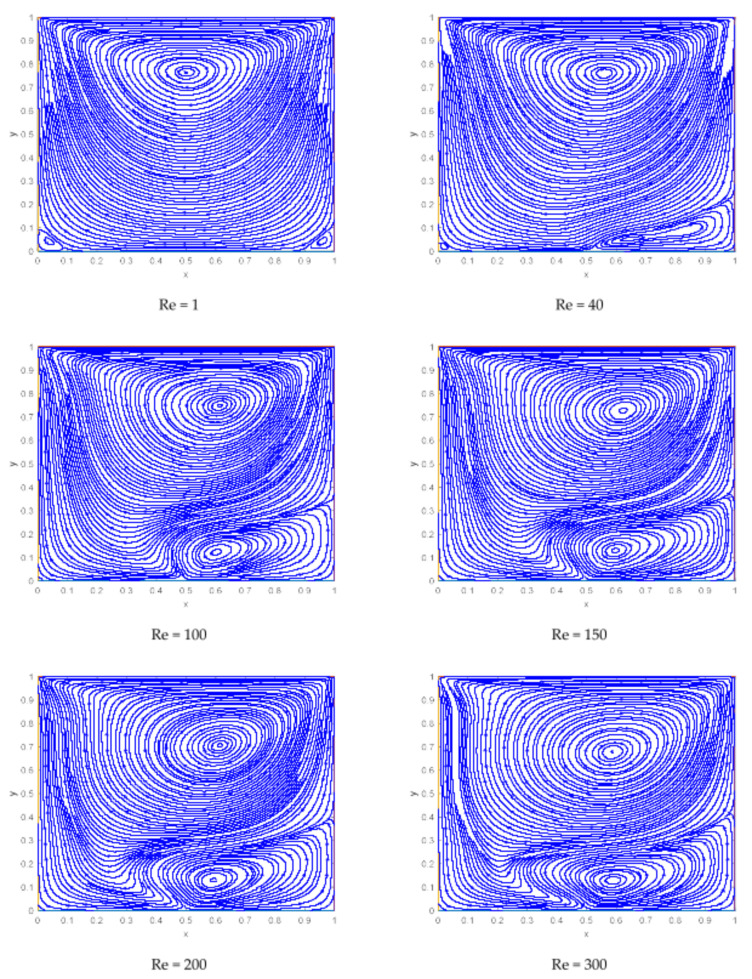
Representation of streamlines for various values of Reynolds numbers.

**Figure 10 materials-15-02761-f010:**
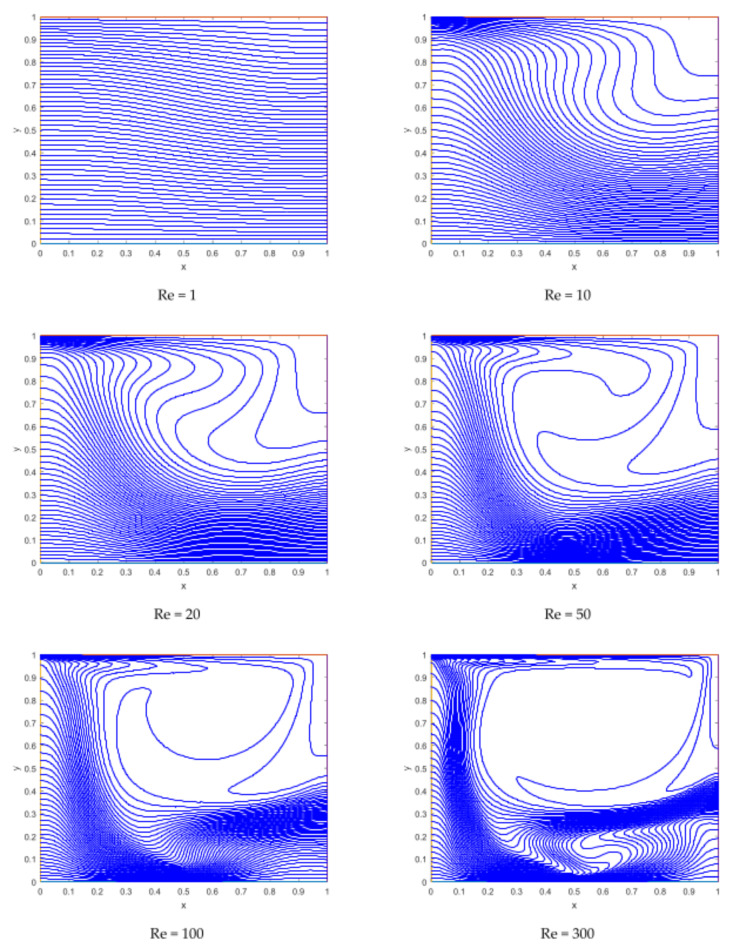
Representation of isotherms for different values of Reynolds numbers.

**Table 1 materials-15-02761-t001:** Prime parameters in Equations (8) and (9).

$\varepsilon =\frac{T_{2}}{T_{2}-T_{1}},$	Dimensionless Temperature number,	Re=Lvoνf,	Reynolds number,
$Mn=\frac{{\bar{\mu }}_{o}{\tilde{H}}_{r}^{2}K\Delta T}{{\tilde{\rho }}_{f}v_{o}{}^{2}},$	Magnetic number,	φ,	Nanoparticle volume fraction,
$\mathrm{Pr}=\frac{{\left(\tilde{\mu }c_{p}\right)}_{f}}{k_{f}},$	Prandtl number,	$Ec=\frac{v_{o}{}^{2}}{\Delta T{\left(c_{p}\right)}_{f}},$	Eckert number.

**Table 2 materials-15-02761-t002:** Average Nusselt number $N_{u}(x)$ comparison with formerly established outcomes.

	$\mathrm{Values}\;\mathrm{of}\;N_{u}(x)\;\mathrm{along}\;\mathrm{the}\;\mathrm{Heated}\;\mathrm{Wall}$	
	$Ra\;=\;10^{4}$	$Ra\;=\;10^{3}$
Chen et al. [33]	2.253	1.119
De Vahl Davis [34]	2.243	1.118
Present Results	2.248	1.118

**Table 3 materials-15-02761-t003:** Thermophysical features of nanoparticles and water.

	$\tilde{\rho }\;\left(kgm^{-3}\right)$	$C_{p}\;\left(Jkg^{-1}K^{-1}\right)$	$k\;\left(Wm^{-1}K^{-1}\right)$	$\beta \;\left(K^{-1}\right)$
Water ($H_{2}O$)	997.1	4179	0.613	40
Alumina(${\mathrm{Al}}_{2}O_{3}$)	3970	765	$21\times 10^{-5}$	$0.85\times 10^{-5}$

**Table 4 materials-15-02761-t004:** Influence of nanoparticle volume fraction ($\tilde{\varphi }$) on physical quantities.

$\tilde{\varphi }$	$Nu_{l}$	$Nu_{u}$	$Cf_{l}$	$\;Cf_{u}$
0.00	1.8959	1.9542	0.3504	20.8050
0.02	1.8410	1.8939	0.3475	20.8043
0.04	1.7742	1.8212	0.3424	20.8011
0.06	1.7010	1.7420	0.3382	20.7989
0.10	1.5499	1.5798	0.3336	20.7975

**Table 5 materials-15-02761-t005:** Influence of Reynolds number (Re) on physical quantities.

Re	$Nu_{l}$	$Nu_{u}$	$Cf_{l}$	$\;Cf_{u}$
1	1.0215	1.0245	0.3300	20.7977
50	3.4323	3.8051	1.0491	21.1033
100	4.7119	5.3072	2.6006	21.8490
150	6.2315	6.8697	3.8441	22.6710
300	9.3855	9.9663	6.2269	24.8867

**Table 6 materials-15-02761-t006:** Impact of the magnetic interaction parameter (Mn) on the physical quantities.

Mn	$Nu_{l}$	$Nu_{u}$	$Cf_{l}$	$\;Cf_{u}$
0	1.8306	1.8831	0.3296	20.8114
200	2.2166	2.2795	6.4334	20.6657
500	2.7604	2.8265	18.2207	20.6879
1000	3.2821	3.3370	35.3971	20.6806
3000	3.9503	3.9574	86.6714	20.8637

## Data Availability

Not applicable.

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
