# Peer review of "Numerical Assessment of Dipole Interaction with the Single-Phase Nanofluid Flow in an Enclosure: A Pseudo-Transient Approach"

_materials, 2022, doi:10.3390/ma15082761_

Round 1
Reviewer 1 Report
Review for Paper Number 1647213 :
This study numerically investigates the effect of magnetic source on nanofluid flow inside a lid-driven square cavity heated from below. Although the effect of “dipole” is mentioned, in literature, there are many similar studies with keywords such as magnetic source and variable magnetic field. Single Phase nanofluid model is adopted.
The paper needs a very careful revision in order to be considered as a publication in the journal.
Some points may listed as follows :
1) In some words, a small change in font size is noticed.
2) “time dependent” should be emphasized in abstract.
3) “Raleigh” should be corrected as “Rayleigh”. “stream-vorticity form” should be written as “stream function-vorticity form”.
4) Why did you chose to use Single Phase Model ?
5) On Problem Figure, T_h and T_c, and also (a,b) are not written.
In the first paragraph of Section 2, it would be better to see which type of nanoparticle is used with which base fluid.
6) Time dependence in Energy Equation 4) is missing. Please check.
7) For a broad sense of readers, a nomenclature would be better.
8) In Eq.(9), time derivative is not seen again. Further, I think in Eqs. (8) and (9), some relations/polynomials depending on solid volume fraction are used. What are these equations ? Are these some experimental equations for dynamical and thermal conductivity of nanofluid ? If so, please clearly state.
9) Subindices should be clearly stated.
10) Eq.(16) is repeated at Eq.(21). In Eq.(16), convection terms are not seen, and where is Prandtl number in this equation ?
11) For iteration, which method (only “direct or iterative method” is written) did you use ? Please mention.
12) Although the keywords involve pseudo-transient approach, it is not noted an emphasize on “pseudo-time” ? What is the advantage or disadvantage of “pseudo time” ?
13) What about electrical conductivity modelling for nanofluid flow in governing equations ? Why does Hartmann number not exist ?
14) Stopping criteria for iteration is not written.
15) The fixed Rayleigh number in numerical results is not mentioned. If it is fixed at Ra=10^4, why not Ra=10^5 ?
16 ) If Eckert number is so small, do you think that the terms with Ec number in governing equations have an importance ?
17) What about delta_t ? The used time increment is not mentioned.
18) Fixing which parameters, grid independence analysis is carried out ? It is not mentioned.
19) In numerical results, explanation for Fig.4 is missing.
20) For explanation of different figures, a new paragraph would be better.
21) Percentage changes related with Table 3 are not clear. From which parameter to which parameter, %X reduction or %X increase occurs ?
Emphasis of these percentage changes may be better in Conclusion.
Author Response
We authors are very thankful to the reviewer for the constructive remarks and suggestions to make the manuscript more reliable and suitable for publication. All queries are well entertained in the revised manuscript and highlighted (coloured) with yellow paint. Please see our responses in attachment file.

Reviewer 2 Report
Review Comments for Manuscript Number: materials-1647213-peer-review-v1
|
Title: |
Numerical Assessment of Dipole Interaction with the Single-Phase Nanofluid Flow in an Enclosure: A Pseudo-Transient Approach |
|
Journal: |
Materials |
The authors present a numerical approach to analyzing the assessment of dipole interacted with single-phase nanofluid flow in an enclosure using the Pseudo-Transient method. The idea is to figure out how much a magnetic field affects nanofluid flow in an enclosure because of a dipole. The authors revealed that Re and magnetic field have a low effect on Nusselt number and skin friction. On the other hand, The Nusselt number rises near the location of the dipole. Additionally, the magnetic field highly affects the high-temperature regions nearby the dipole, while newly generated vortices rotate in different directions. The research idea is interesting. However, I’m not sure if we can rely on it when changing the boundary conditions. It didn’t consider the environmental parameters that may affect all the results. I urge the authors to support their outcomes with some theoretical explanations. Additionally, here are some points the authors should work on:
- Major English revision is needed. There is no need for ambiguous wordings. Just make it simple and readable.
- In the introduction, Lines 32-33: “Therefore, the amalgamation of the base fluid in the tiny sized solid constituents would improve the heat transfer rate.” Please add your reference for this conclusion.
- In the introduction, Lines 39-44, Are the referred applications of the nanofluids are already applied in the industry? If not, please highlight that. (There are plenty id studies about using nanofluids, however, I didn’t see the real use of them in the industry).
- The flowing process in the introduction is not smooth. Additionally, abundant irrelevant details are included. The authors should concise it and directly to their topic after giving a comprehensive overview about the general topic (nanofluids). The introduction must be carefully revised.
- In the Material and Methods, Lines 157-159, This study considered only study conditions for temperatures and magnetic fields. Am I right? If so, the authors should make this clear in this study since changing the boundary conditions (BCs) change all the theories provided in this article.
- I noticed that the authors removed the pressure term, is this an assumption?
- In the Material and Methods, there is no need for using “Mow, we ….”, Now is the time …”. Revise them!
- Are there other BCs could affect the overall system?
- Line 198, the subsection should be 3.1.1. Same thing for Line 210, 3.1.2.
- I don’t get the information presented in Lines 199-202. Explain them, please.
- Sub-section 3.2, provide a small introduction and explaination about your flowchart.
- Lines 259-264 plus to all the paragraphs in the results and discussions sections require further explanations (explain why).
- The conclusions should have a small intro as in the abstract (just about the main point of this research).
- In essence, it is good that the authors validated their results as in Table 1. However, they need to support their outcomes with some theoretical explanations.
Author Response

(The authors gave the same response as above.)

Round 2
Reviewer 2 Report
Review Comments for Manuscript Number: materials-1647213-peer-review-v2
|
Title: |
Numerical Assessment of Dipole Interaction with the Single-Phase Nanofluid Flow in an Enclosure: A Pseudo-Transient Approach |
|
Journal: |
Materials |
The authors' response is not straightforward to my comments. However, I see the efforts in their revised manuscript. In general, I’m satisfied with all of their responses. Thus, I recommend it for publication after a minor revision.